# Short-Term Dietary Restriction Potentiates an Anti-Inflammatory Circulating Mucosal-Associated Invariant T-Cell Response

**DOI:** 10.3390/nu16081245

**Published:** 2024-04-22

**Authors:** Brian Fazzone, Erik M. Anderson, Jared M. Rozowsky, Xuanxuan Yu, Kerri A. O’Malley, Scott Robinson, Salvatore T. Scali, Guoshuai Cai, Scott A. Berceli

**Affiliations:** 1Department of Surgery, Division of Vascular Surgery and Endovascular Therapy, University of Florida, Gainesville, FL 32611, USA; brian.fazzone@ufl.edu (B.F.); erik.anderson@surgery.ufl.edu (E.M.A.); kerri.omalley@surgery.ufl.edu (K.A.O.); scott.robinson@surgery.ufl.edu (S.R.); salvatore.scali@surgery.ufl.edu (S.T.S.); 2Department of Epidemiology and Biostatistics, Arnold School of Public Health, University of South Carolina, Columbia, SC 29208, USA; xuanxuan@email.sc.edu; 3Malcom Randall Veteran Affairs Medical Center, Gainesville, FL 32608, USA; 4Department of Environmental Health Sciences, Arnold School of Public Health, University of South Carolina, Columbia, SC 29208, USA; guoshuai.cai@surgery.ufl.edu

**Keywords:** short-term dietary restriction, mucosal-associated invariant T-cells, MAITs, inflammation, immune response

## Abstract

Short-term protein-calorie dietary restriction (StDR) is a promising preoperative strategy for modulating postoperative inflammation. We have previously shown marked gut microbial activity during StDR, but relationships between StDR, the gut microbiome, and systemic immunity remain poorly understood. Mucosal-associated invariant T-cells (MAITs) are enriched on mucosal surfaces and in circulation, bridge innate and adaptive immunity, are sensitive to gut microbial changes, and may mediate systemic responses to StDR. Herein, we characterized the MAIT transcriptomic response to StDR using single-cell RNA sequencing of human PBMCs and evaluated gut microbial species-level changes through sequencing of stool samples. Healthy volunteers underwent 4 days of DR during which blood and stool samples were collected before, during, and after DR. MAITs composed 2.4% of PBMCs. More MAIT genes were differentially downregulated during DR, particularly genes associated with MAIT activation (CD69), regulation of pro-inflammatory signaling (IL1, IL6, IL10, TNFα), and T-cell co-stimulation (CD40/CD40L, CD28), whereas genes associated with anti-inflammatory IL10 signaling were upregulated. Stool analysis showed a decreased abundance of multiple MAIT-stimulating *Bacteroides* species during DR. The analyses suggest that StDR potentiates an anti-inflammatory MAIT immunophenotype through modulation of TCR-dependent signaling, potentially secondary to gut microbial species-level changes.

## 1. Introduction

Short-term dietary restriction (StDR) without malnutrition is an intriguing preoperative conditioning strategy for modulating the postoperative immune response [1,2,3]. Broadly, attenuation of proinflammatory signaling and production of anti-inflammatory chemokines are proposed mechanisms for its benefits [1,4,5,6]. Multiple, small, randomized controlled trials have demonstrated StDR safety and feasibility, yet mechanistic insights into its modulation of the human immune system remain limited [7,8,9]. The gut microbiome influences systemic immunity, and its modification represents an important area for translational therapeutic development [10,11,12,13,14,15]. Our group has previously shown a relative gut microbial dysbiosis after 4 days of protein-calorie restriction with a rapid return to baseline after resuming a normal diet [16]. However, characterization of the gut microbiome alone after StDR does not describe its influence on systemic immunity. 

Mucosal-associated invariant T-cells (MAITs) are a subset of T-cells involved in early immunity against infection, providing a more rapid response to pathogens than conventional MHC-restricted T cells [17,18]. MAITs are defined by their invariant T-cell receptor (TCR) complex, which is restricted to recognition of MHC class-I-related (“MR1”) ligands [17,18,19]. Few MAIT TCR ligands are known, but the most potent ligands are metabolites of bacterial riboflavin (vitamin B2) biosynthesis and folate (vitamin B9) degradation [17,18,19,20,21]. Furthermore, riboflavin-producing bacteria belonging to the *Bacteriodetes* and *Proteobacteria* phyla are the most potent MAIT stimulators in vitro [22]. MAITs are activated in a TCR-dependent or -independent manner [20]. TCR-dependent signaling requires ligand presentation to the MAIT TCR and co-stimulation by antigen-presenting cells (APCs) (Figure 1), whereas TCR-independent stimulation relies upon multiple signaling chemokines [20,23,24].

Driven by their enrichment on mucosal surfaces and bacterial metabolite stimulations, MAIT function is sensitive to alterations in gut microbial metabolites and represents an important link between the microbiome and systemic immunity [21,22,23,24,25]. However, the influence of diet on the functional phenotype of MAITs has not been described. We hypothesized that StDR would induce an anti-inflammatory circulating MAIT phenotype given that StDR rapidly affects the gut microbial composition and that antigenic MAIT stimulation is sensitive to changes in bacterial metabolites. 

## 2. Materials and Methods

### 2.1. Patient Selection, Dietary Restriction, and Sample Collection

Healthy, young adult males and females free of chronic medical conditions were recruited to participate in this study. Informed consent was obtained from all subjects before study participation, and this study was approved by the University of Florida’s institutional review board (UF IRB#201900988). Participant demographics and medical histories were self-reported. Blood and stool samples were collected prior to dietary intervention (“Baseline”). A liquid diet was initiated targeting 30% caloric and 70% protein restriction, assuming a pre-intervention protein caloric intake of 15%. The caloric allowance was individualized based on participants’ total caloric requirements. The Paffenbarger physical activity questionnaire was used to estimate each participant’s total caloric requirement based on their resting energy expenditure (REE) and daily activity [26,27]. The Mifflin St. Jeor equation (Equation (1)) was used to calculate REE [28]: REE = 9.99 × weight (kg) + 6.25 × height (cm) − 4.92 × age (yrs) + 166 × sex (male = 1, female = 0) − 161(1)

No activity restrictions were imposed during the study period. Participants were instructed to only consume powder shakes (Scandishake^®^ Mix, Nutricia Advanced Medical Nutrition, Utrecht, The Netherlands) mixed with almond milk for 4 days of dietary restriction. Scandishake^®^ Mix is a low-protein mixture containing 491 Kcal, 4% protein, 42% carbohydrates, and 44% fat. This mixture was chosen given its low protein content, thus feasibility in achieving 70% protein restriction. Dietary adherence was monitored during follow-up appointments and with a MealLogger mobile application shared with the study staff. Participants were allowed to drink water ad libitum. Blood samples were collected after two (“Day 2”) and four (“Day 4”) days of dietary restriction and three days after resuming a normal diet (“Day 7”). Stool samples were collected on days 4 and 7.

### 2.2. Peripheral Blood Mononuclear Cell Isolation from Blood

At least 4 mL of blood was collected via venipuncture in EDTA-containing blood tubes for each blood sample. Samples were processed within 30 min of collection. A 1× lysing solution was prepared with a 10:1 mixture of double distilled water and 10× red blood cell lysis solution (Miltenyl Biotec; Cologne, Germany). A total of 4 mL of whole blood was added to the 1× lysing solution and incubated at room temperature for 10 min. The homogenized solution was then centrifuged at 300× *g* for 10 min. The supernatant was removed, and the pellet was resuspended in 10 mL of 1× lysing solution for a 2nd lysis step. The solution was centrifuged again at 300× *g* for 10 min, and the resultant pellet was then resuspended in 1× Hank’s Balanced Salt Solution (Thermo Fisher Scientific corporation; Waltham, MA, USA). After washing, the solution was centrifuged at 300× *g* for 10 min. The final pellet was resuspended in a frozen medium containing 5:1 fetal bovine serum (Thermo Fisher Scientific corporation; Waltham, MA, USA) and Dimethyl Sulfoxide [DMSO] (Thermo Fisher Scientific corporation; Waltham, MA, USA). The isolated cellular solution was placed in a 2 mL cryotube and in liquid nitrogen for long-term storage. 

### 2.3. Blood Sample Thawing for Single-Cell Workflow

Samples were rapidly thawed using a warm water bath set to 38 degrees Celsius. Thawed samples were serially diluted with warm RPMI 1640 (Corning Incorporated; New York, NY, USA) added in a drop-wise manner. The diluted samples were centrifuged at 300× *g* for 10 min. The supernatant was removed, and the pellet was resuspended in a phosphate-buffered saline/0.05% BSA solution (Thermo Fisher Scientific corporation; Waltham, MA, USA). The solution was filtered through a 70 um filter to remove large debris and centrifuged at 300× *g* for 10 min. The supernatant was again removed, and the pellet was resuspended in PBS/0.05% BSA. The solution was filtered through a 40 um cell strainer (Thermo Fisher Scientific corporation; Waltham, MA, USA) to remove micro-debris. Cellular viability was assessed using Trypan blue staining and a Countess II cell counter (Thermo Fisher Scientific corporation; Waltham, MA, USA). Samples did not proceed to library generation if cellular viability was <90%. 

### 2.4. Library Generation and Sequencing

Libraries were generated per manufacturer instructions using the 10× Chromium Single-Cell v3.1 protocol (10X Genomics; Pleasanton, CA, USA). Briefly, thawed cells were loaded into the 10× chip targeting a cellular recovery of 3000 cells per sample. The chip was loaded into the Chromium controller which utilizes microfluidics to capture individual cells within gel beads in emulsion (GEMs) containing uniquely barcoded gel beads and reagents for reverse transcription. Cells were lysed within GEMs, and full-length DNA transcripts were created from mRNA transcripts. After purification, cDNA transcripts were amplified through a series of PCR reactions. Quality analysis of the final library was performed before sequencing using the Agilent TapeStation (Agilent Technologies; Santa Clara, CA, USA). Libraries were sent to Quick Biology Inc. (Pasadena, CA, USA) for sequencing. Samples were sequenced using the NovaSeq 6000 system (Illumina; San Diego, CA, USA) with paired-end reads and 100 million pairs per sample. 

### 2.5. Data Preprocessing and Filtering

The raw data in the form of FASQC files were uploaded directly to the Cell Ranger pipeline (10X Genomics; Pleasanton, CA, USA) for demultiplexing, genome alignment (GRCh38), and quality analysis. The feature-unique molecular identifier (UMI) matrix obtained from Cell Ranger was uploaded to R studio using the R statistical software package (V 4.1.1., the R Foundation for Statistical Computer, Vienna, Austria) for downstream processing. Quality control, normalization, dimensional reduction, and clustering were performed using the Seurat package (V 4.3.0) for R. Cells with mitochondrial percentages > 10% or features < 200 or >7500 were filtered out to exclude dead cells, empty droplets, or multiplets. Log normalization was performed, and the top 2000 most variable genes were used for principal component analysis (PCA). The top 20 principal components were chosen for t-distributed stochastic neighbor embedding (t-SNE) and uniform manifold approximation and projection (UMAP). 

### 2.6. Annotation and MAIT Identification

Individual clusters were identified using a combination of manual annotation and automated annotation with ScType. ScType is an unsupervised, automated, reference-based annotation tool used for cluster identification [29]. ScType relies on a large database of established cell markers and utilizes positive and negative markers of gene expression to annotate cells. MAITs were identified using KLRB1 (encodes killer cell lectin-like receptor B1) and SLC4A10 (encodes solute carrier family 4 member 10) [23,30,31] (Figure 2C). SLC4A10 is a highly specific marker for MAIT cells, having a Tau specificity score of 0.93 [32]. A new Seurat object for differential expression analysis was created using the annotated MAITs. The total number of MAITs, MAITs/condition, and MAITs/sample/condition was reported as absolute values and percentages. Values are reported as means ± standard deviation. A one-way ANOVA was used to compare MAITs/sample between conditions.

### 2.7. Differential Expression and MAIT Activation

Differentially expressed genes (DEGs) between time points were detected by a linear mixed model, which accounts for within-subject variability and within-group variability (Equation (2)). In this model, the time point was modeled as a dummy variable and day 0 was treated as the baseline. Specifically, the linear mixed model for gene g, subject i, cell j can be expressed as
(2)$$y_{ijg}=\beta_{0}+\beta_{1}*{time2}_{ij}+\beta_{2}*{time4}_{ij}+\beta_{3}*{time7}_{ij}+\alpha_{i}+\varepsilon_{ijg}\;\alpha_{i}\sim N\left(0,\;\tau^{2}\right),\;\varepsilon_{ijg}\sim N(0,\omega_{ijg}\sigma_{g}^{2})$$
where $\beta_{1},\;\beta_{2},\beta_{3}$ denote the mean changes in gene expression comparing day 2, day 4, and day 7 to the baseline time point, respectively. $\alpha_{i}$ represents the random effect corresponding to subject i with $\tau^{2}$ measuring the variance of random subject effect. $\varepsilon_{ijg}$ is the random error in gene g with variance $\sigma_{g}^{2}$. To unlock linear model analysis for scRNA-seq data, limma-voom precision weight for the mean–variance dependency [33] and ZINB-WaVE weight for dropout probability [34] were multiplicatively introduced into the weight $\omega_{ijg}$ in the model. The analysis was based on the limma framework [33], and its moderated t-statistics were used for obtaining *p*-values. Further, false discovery rate (FDR) correction was performed to adjust for multiple comparisons. An FDR-adjusted *p*-value < 0.01 was considered statistically significant.

Differentially expressed genes were identified by the following criteria: expressed in ≥5% of MAITs, an absolute log-fold change (LogFc) from baseline ≥ 0.4, and an FDR-adjusted *p*-value < 0.01. Functional enrichment analysis was performed using differentially expressed genes. Differential expression of the specific markers CD69 (encodes human transmembrane *C*-type lectin protein), CD25 (encodes interleukin-2 receptor alpha chain), and HLA-DR (encodes MHC class II cell surface receptor) was used to evaluate MAIT activation. CD69 is considered the earliest marker of T-cell activation, is transcriptionally expressed rapidly after MAIT cell activation, correlates well with patterns of inflammatory disease, and may be an important regulator of the intestinal immune system [23,35,36]. CD25 and HLA-DR are well-characterized late lymphocyte activation markers that can be upregulated specifically after MAIT cell activation [37,38,39]. 

### 2.8. Functional Enrichment Analysis

Functional enrichment analysis was performed using Qiagen Ingenuity Pathway Analysis (IPA) software v.01.23.01 (Qiagen; Hilden, Germany) [40]. IPA utilizes statistical analysis and computational modeling combined with an extensively curated database of genetic knowledge for functional pathways, upstream regulators, and network analyses. The primary analysis was based on the criteria consistent with the above analysis, an absolute LogFc cut-off of 0.4 to define down- and upregulated genes and false detection rate (FDR) < 0.1. Both direct and indirect relationships were considered in the pathway analysis. The Ingenuity Knowledge Base was used as a reference set. All comparisons were made between the dietary condition (“Day 2”, “Day 4”, “Day 7”) and baseline.

Significant canonical pathways were identified using a Fisher’s exact test with an FDR-corrected *p*-value of overlap threshold for significance set to <0.05. The Fisher’s exact test tests overlap between the input genes from the experimental dataset and genes associated with known signaling pathways expertly curated from the literature. Pathways are considered enriched if the input genes have significant overlap with known pathway genes. Pathways associated with apoptosis, cellular immune response, cellular stress and injury, cytokine signaling, nuclear receptor signaling, and pathogen-influenced signaling were evaluated. The Qiagen software calculates Z-scores to generate pathway activation predictions based on which genes are upregulated or downregulated in the dataset. Negative Z-scores indicate pathway inhibition, whereas positive Z-scores indicate pathway activation. Pathways were considered significantly inhibited or activated if their Z-score was less than −2.0 or greater than 2.0.

Upstream regulator molecules were identified using the IPA software. Regulators are any molecules known to drive the differential expression changes observed in the input dataset. Statistical testing for regulator analysis is similar to the techniques performed for canonical pathway analysis but focused on curated gene–regulator relationships. Significance is based on the degree of overlap between differentially expressed genes in the input dataset and known molecules within a particular regulatory network. Significant regulators were identified by an FDR-adjusted *p*-value of overlap < 0.01. Regulators were significantly activated (positive Z-score) or inhibited (negative Z-score) if their absolute Z-score was greater than 2.0. All regulators associated with genes, RNA, and proteins were evaluated.

### 2.9. Stool Sample Collection, Processing, Filtering, and Sequencing

Stool samples were collected by the patients using a fecal collection kit (Zymo Research, Irvine, CA, USA), and samples were stored at −20 degrees Celsius until processing. As previously described [16], the ZymoBiomics MagBead DNA kit (Zymo Research, Irvine, CA, USA) was used to isolate DNA from thawed stool samples and shotgun metagenomic sequencing (Zymo Research, Irvine, CA, USA) was performed. Whole-genome sequencing libraries were created with the Nextera DNA Flex Library Prep Kit (Illumina, San Diego, CA, USA), pooled in equal abundance, and sequenced with NovaSeq (Illumina, San Diego, CA, USA). Low-quality reads were removed, and metagenomic compositional profiling and abundance analysis was performed with Centrifuge [41] and functional profiling was performed using HUMAnN2.

### 2.10. Microbiome Taxonomic and Functional Pathway Analysis

We have previously described our methods for taxonomic and functional pathway analysis [16]. Briefly, low-prevalence species were filtered from the dataset using a machine learning algorithm to identify an ideal prevalence filtering cutoff. Species present in at least 60% of the samples were included in downstream analysis. Conditional variance was assessed using principal component analysis, and species-level differential abundance analysis was performed using ALDEx2. Specifically, Monte Carlo sampling from a Dirichlet distribution, followed by center-log-ratio transformation and Kruskal–Wallis significance testing with false discovery rate (FDR) correction, was used to analyze species differences between all three conditions, followed by Welch *t*-testing with correction and effect size calculations to assess directional changes between baseline and day 4 conditions. Heatmaps were created with hierarchical clustering based on Euclidean distances with center-log-ratio transformation. Like taxonomic analysis, prevalence filtering was performed for functional pathways, using 80% as the ideal prevalence threshold. Conditional differences were again evaluated using principal component analysis, followed by individual pathway analysis. Normalized reads were compared between the three conditions using ANOVA testing controlled for FDR. Heatmaps were similarly created for pathway functional categories.

## 3. Results

### 3.1. Participant Demographics

Seven participants completed the dietary intervention and provided samples for analysis. Table 1 depicts their characteristics. All participants were healthy nonsmokers. Four participants were male, and three were female. Five were White, one was Hispanic, and one was Asian. The average age was 25.3 (±3.8) years, and their average BMI was 25.7 (±4.0). All participants provided blood samples for each time point (n = 28 samples). All provided baseline and day 7 stool samples. Two were unable to produce a day 4 stool sample (n = 19 samples).

### 3.2. Filtering and Annotation

A total of 65,372 cells were isolated across all samples with a mean of 85,595 reads per cell. A total of 57,064 cells remained after applying filtering cut-offs. A total of 16 cell clusters were identified. Figure 2A shows the annotated UMAP of cell clusters. MAITs represented 1351 (2.4%) of isolated circulating PBMCs, and MAIT proportions remained consistent throughout the study period (Figure 2D). The average number of MAITs per sample was 48.2 ± 26.6. A total of 342 (2.4%) MAITs were present at baseline compared to 322 (2.3%) on day 2, 350 (2.3%) on day 4, and 337 (2.4%) on day 7. We did not observe a significant difference between the number of MAITs/sample by condition (*p* = 0.991).

### 3.3. Differential Expression and MAIT Activation

Compared to the baseline, 232, 172, and 145 genes were differentially expressed on days 2, 4, and 7 of the diet, respectively. A total of 93 genes were upregulated on day 2, 24 on day 4, and 85 on day 7. A total of 139 genes were downregulated on day 2, 148 on day 4, and 60 on day 7. Figure 3 shows volcano plots for the most differentially expressed genes by condition. A total of 60% and 84% of the genes were significantly disproportionally downregulated on days 2 and 4 compared to 40% on day 7 (Chi-square-test *p* < 0.001). The top 10 most downregulated and upregulated genes per condition are shown in Appendix A. The late activation-specific genes CD25 (LogFc 0.17, 0.04, 0.15 on days 2, 4, and 7, respectively) and HLA-DR (LogFc 0.01, 0.18, 0.06 on days 2, 4, and 7, respectively) were not differentially expressed at any time point. CD69 was differentially downregulated on days 2 (LogFc = −0.66, *p* < 0.001) and 4 (LogFc = −0.49, *p* < 0.001) compared to baseline, but not on day 7 (LogFc = −0.38).

### 3.4. Canonical Pathways

A total of 176 and 160 pathways were functionally enriched on days 2 and 4 of the diet. A total of 11 pathways were enriched on day 7 (Appendix A). A total of 71 (40%) and 64 (40%) pathways were significantly (absolute Z-score ≥ 2.0) inhibited or activated on days 2 and 4, respectively. Of the significantly changed pathways, sixty-seven (94%) were inhibited and four (6%) were activated on day 2, and fifty-seven (89%) were inhibited and seven (11%) were activated on day 4. There was considerable overlap in enriched pathways and pathway activation states between days 2 and 4. A total of 39/64 (61%) of the significantly inhibited or activated pathways day 4 pathways overlapped with day 2.

Figure 4 depicts select enriched canonical pathways on days 2 and 4 compared to baseline. TNFR1 and TNFR2 signaling were among the most enriched pathways on days 2 and 4. Both had evidence of inhibition on both days, with TNFR1 being significantly inhibited on day 2 (Z = −2.131). Acute phase response (Z = −2.236, −2.000), IL6 (Z = −2.236, −2.449), iNOS (Z = −2.000, −2.000), and IL1 (Z = −2.000, −2.236) signaling pathways were enriched and significantly inhibited on days 2 and 4. IL17 signaling was enriched and significantly inhibited on day 4 (Z = −2.000). IL10 signaling was enriched on days 2 and 4 and was the most activated pathway on day 2 (Z = 2.530). It remained activated on day 4 (Z = 1.890). The pathogenic induced cytokine storm signaling pathway was among the most inhibited on day 4 (−3.000). Multiple T-cell costimulatory pathways were enriched, and all showed evidence of inhibition during dietary intervention (CD28 and CD40 signaling on day 2 [Z = −1.342 and −0.447], CD40 signaling on day 4 [Z = −0.447]). TCR signaling was enriched on day 2 with evidence of inhibition (Z = −1.897). Immunogenic cell death (Z = −2.000) and death receptor signaling (Z = −1.414) pathways were inhibited on both days, but apoptosis signaling had evidence of activation on day 4 (Z = 0.816). 

PPAR signaling was enriched and had evidence of activation on days 2 and 4 (Z = 1.633 and 2.000, respectively). PPAR/RXRα activation signaling was enriched with evidence of activation on day 2 (Z = 1.342). No day 7 pathways were significantly activated or inhibited. Death receptor signaling was the only pathway with evidence of activation (Z = 1.000). There were minimal changes in pathways associated with MAIT TCR-independent signaling on any day. There was no enrichment of IL18 signaling on days 2 and 4, IL15 signaling on day 2, or IL7 signaling on day 4. IL12 signaling was enriched on days 2 and 4 with evidence of inhibition (Z = −0.333 and −0.816, respectively). IL15 production was the only TCR-independent associated pathway with a significant functional change on day 4 (Z = −2.000). Generally, pathway activation states on day 2 of the diet were like day 4 (Figure 4). Directional changes were 100% congruent among significantly changed pathways (absolute Z-score ≥ 2.0) between days 2 and 4. 

### 3.5. Regulators

Seventy-one, one hundred nineteen, and six genes, RNA, or protein regulators with a significantly changed activation state were identified. A total of 55% (n = 39) of day 2 regulators were also identified on day 4. A total of 57 (80%) and 101 (85%) of day 2 and 4 regulators were inhibited. There was 100% congruence in activation states among overlapping regulators on days 2 and 4. CD3 (encodes T-cell receptor) and the costimulatory regulators CD40 and CD40LG were inhibited on days 2 and 4. Multiple major pro-inflammatory regulators (TNF, IL1, IL6) and IL2 were inhibited on days 2 and 4. IFNγ was significantly inhibited on day 4 (−2.469). Table 2 depicts the top 10 most significant regulatory cytokines and complexes on days 2 and 4 and each regulator’s functional significance. The most inhibited regulators were involved in the TCR complex, antigen-presenting cell co-stimulation, and proinflammatory signaling. The entire list of significant regulators for each condition is provided in the Appendix A.

### 3.6. Species Abundance Analysis

A total of 758 total species were identified in participant stool samples. A total of 168 (22.1%) were present after using the 60% prevalence filtering cut-off. A total of 22 total species were significantly changed between conditions (FDR-adjusted *p* < 0.10). Species-level abundances are most similar between the baseline and day 7 conditions compared to day 4, with unsupervised clustering showing grouping by condition rather than participant (Appendix A). Figure 5 depicts the significant species with an absolute effect size > 1.0 between baseline and day 4. StDR decreased the relative abundances of multiple *Bacteroides* species: *B. stercoris* (Effect size = −4.03), *B. dorei* (Effect size = −2.95), *B. fragilis* (Effect size = −1.34), *B. galacturonicus* (Effect size = −1.29), and increased the abundances of multiple *Ruminococcus* and *Firmicutes species*: *R. torques* (Effect size = 6.32), *R. faecis* (Effect size = 2.14), *Firmicutes* bacterium CAG83 (Effect size = 4.89), and *Firmicutes* bacterium CAG145 (Effect size = 1.73). 

### 3.7. Microbial Functional Pathway Analysis

A total of 345 unique MetaCyc pathways were identified. A total of 177 (51.3%) pathways remained after using a prevalence cut-off of 80%. A total of 34 pathways were significantly changed between conditions (FDR-adjusted *p* < 0.05). Like the abundance analysis, baseline and day 7 functional pathways were more like each other than day 4 (Appendix A) and functional pathways grouped well by condition after unsupervised clustering. MetaCyc pathways are subcategorized based on their metabolic function and associated metabolites. Of the significantly changed pathways between conditions, nineteen (56%) were biosynthetic pathways, twelve (35%) were degradation pathways, and three (9%) were precursor metabolites and energy pathways (Figure 6). A total of 29/34 (85%) species changed by day 4 compared to baseline had reciprocal changes by day 7. A total of 11 (58%) biosynthetic pathways decreased at day 4 compared to baseline. Notably, the pathway for bacterial flavin biosynthesis was decreased and the super pathways for tetrahydrofolate biosynthesis and salvage were both increased on day 4. 

## 4. Discussion

StDR potentiated an anti-inflammatory MAIT immunophenotype characterized by decreased pro-inflammatory TNFα, IL1, IL6, IL17, and IFNγ and increased anti-inflammatory IL10 signaling. Inhibition of the TCR complex, inhibition of APC co-stimulation, and downregulation of the CD69 activation marker suggests modulation of TCR-dependent signaling during StDR as a potential mechanism for the observed effects. This is further supported by gut microbial analysis, which showed multiple *Bacteroides* species among the most decreased species during StDR and favorable changes in bacterial biosynthetic pathways known to produce MAIT-modulating metabolites. Lastly, PPAR signaling was activated during dietary restriction, providing another possible mechanistic link between dietary alteration and the systemic immune response. This is the first study to characterize MAIT transcriptomic responses to StDR in concert with gut microbial changes, identify MAITs as a potential therapeutic target for preoperative StDR preconditioning, and provide transcriptomic evidence for the beneficial immune effects of StDR. 

MAITs exhibit a pro-inflammatory response to riboflavin-producing bacteria dominated by IL17, IFNγ, and TNFα secretion. Circulating MAITs in obesity and metabolic disease are characterized by increased IL2, granzyme B, IL17, IFNγ, and TNFα, and less IL10 compared to healthy controls [17,42,43,44]. Caloric restriction in mouse models has shown a MAIT response characterized by TReg polarization with downregulation of TH17 signaling, resulting in improved stroke outcomes through attenuation of neuroinflammation [45,46]. We observed a similar trend in the MAIT response to StDR characterized by decreased TNFα and increased IL10 signaling with strong inhibition of the major pro-inflammatory IL1, IL6, and iNOS signaling pathways. This trend was consistent throughout the intervention, returning to baseline after diet cessation. Overall, we show a rapid and sustained effect of StDR on systemic MAITs. The clinical benefits of these short-term changes require further exploration; however, long-term dietary restriction in patients with non-alcoholic fatty liver disease resulted in a decrease in intrahepatic MAIT CD69 expression and improvements in inflammatory-mediated hepatic steatosis [47].

MAITs are activated in a TCR-dependent manner by byproducts of bacterial riboflavin biosynthesis presented on MR1 ligands by APCs. APCs co-stimulate MAITs through CD28/CD80 and CD40/CD40L interactions, synergistically increasing the TCR-dependent response and rapidly increasing CD69 expression [38]. TCR-independent activation relies on IL7, IL12, IL15, and IL18 cytokine binding to MAIT receptors and is generally preceded by viral or general inflammatory stimuli [17]. There was no pattern to suggest a coordinated TCR-independent response to StDR; however, we observed decreased activity of CD40, T Cell Receptor, and CD28 signaling during StDR. Regulator analysis further identified CD40, CD28, and the TCR among the most inhibited regulatory complexes. It is unclear whether these trends are secondary to inhibition of the MAIT TCR complex, decreased antigenic MAIT stimulation, or a combination of both; however, we suspect StDR may have the altered the microbiome, potentially producing favorable MAIT modulating metabolites.

Bacteria belonging to the *Bacteroidetes* and *Proteobacteria* phyla are among the most potent MAIT stimulators in vitro [22]. Our previous work showed that 4 days of StDR induced species-level abundance changes that returned to a predietary state 3 days after dietary cessation [16]. Well-established relationships between gut microbial composition and dietary restriction remain elusive given variations in study design, diet selection, and duration of intervention [48]. Caloric restriction models in mice ranging from 2 weeks to 45 days have shown increases in the *Bacteroidetes* phylum, whereas a 3-week hypocaloric hyperproteic diet in humans decreased multiple *Bacteroides* species [49,50]. Our previous work showed a decrease in multiple *Bacteroides* species during 4 days of dietary restriction and this observation was redemonstrated herein. We showed that StDR rapidly alters the gut microbiome, suggesting utility as a perioperative intervention. However, the species-specific effects, ideal dietary intervention, and duration of intervention require further study.

MAIT TCR-dependent activation relies exclusively on metabolites of bacterial riboflavin biosynthesis [17,18,19,20,21]. Transcriptomic profiling of *E. coli*-activated MAITs has revealed distinct pathway changes associated with increased IL1, IL6, TNFR1/R2, IL2, CD40, CD28, and Th1/Th17 signaling pathways [51]. We further observed decreased bacterial ribofvain biosynthesis and increased activity in folate degradation pathways during StDR, potentially resulting in the production of MAIT-inhibiting byproducts [17,18,19,20,21]. It is possible that the observed species-level changes resulted in favorable microbiome metabolite profiles, thereby modulating MAIT behavior. However, a high-resolution analysis of stool metabolites via mass spectroscopy has not been performed and should be the work of future research.

PPAR signaling, particularly PPARα, was one of the few activated pathways during dietary restriction. PPAR receptors are widely expressed on T-cells and have been shown to regulate T-cell activation and differentiation states in response to fatty acid metabolites [52]. PPARα target genes are important regulators of fatty acid oxidation and ketogenesis that downregulate many pro-inflammatory and acute phase response signaling pathways upon activation [53]. Indeed, caloric restriction induces the production of PPARα ligands, leading to a favorable immunoprofile and translating to numerous health benefits [54]. MAIT responses to PPAR agonism are not well-characterized, but this study highlights PPAR modulation as a potentially important mechanism for diet-induced MAIT modulation.

## 5. Limitations

This study has several limitations. The diet was chosen for its low protein content and thus feasibility in achieving 70% protein and 30% caloric restriction. This specific diet is not supplemented with essential amino acids and is relatively high in carbohydrates. The ideal diet for StDR is unknown, but likely involves not only protein and caloric restriction, but also supplementation of essential amino acids and vitamins to promote an ideal gut microbiota composition. Moreover, the observations herein could be secondary to the diet itself and not necessarily from protein or calorie restriction. This study lacks a description of other peripheral immune cell responses to StDR. For example, understanding APC response to StDR could provide support for the mechanistic benefits of StDR on MAIT expression. A comprehensive profiling of the entire PBMC immunophenotype to StDR is the target of future works. The study population was homogenous and the results in healthy, young participants may not be generalizable to other patients, particularly those with multiple confounding comorbidities. Lastly, many of the findings in this study require further validation with cell culture, animal, or translational models.

## 6. Conclusions

StDR induced an anti-inflammatory immunophenotype in circulating MAITs. The MAIT response was rapid and returned to baseline after three days of a normal diet. Cytokine profiling of MAITs during StDR indicated polarization towards an anti-inflammatory TReg-like phenotype and away from the proinflammatory TH1/TH17-like phenotype. Analysis of key pathways and regulators indicates that StDR may modulate systemic MAITs through alterations in TCR-dependent signaling, potentially as a result of microbiome compositional changes during dietary restriction. 

## Figures and Tables

**Figure 1 nutrients-16-01245-f001:**
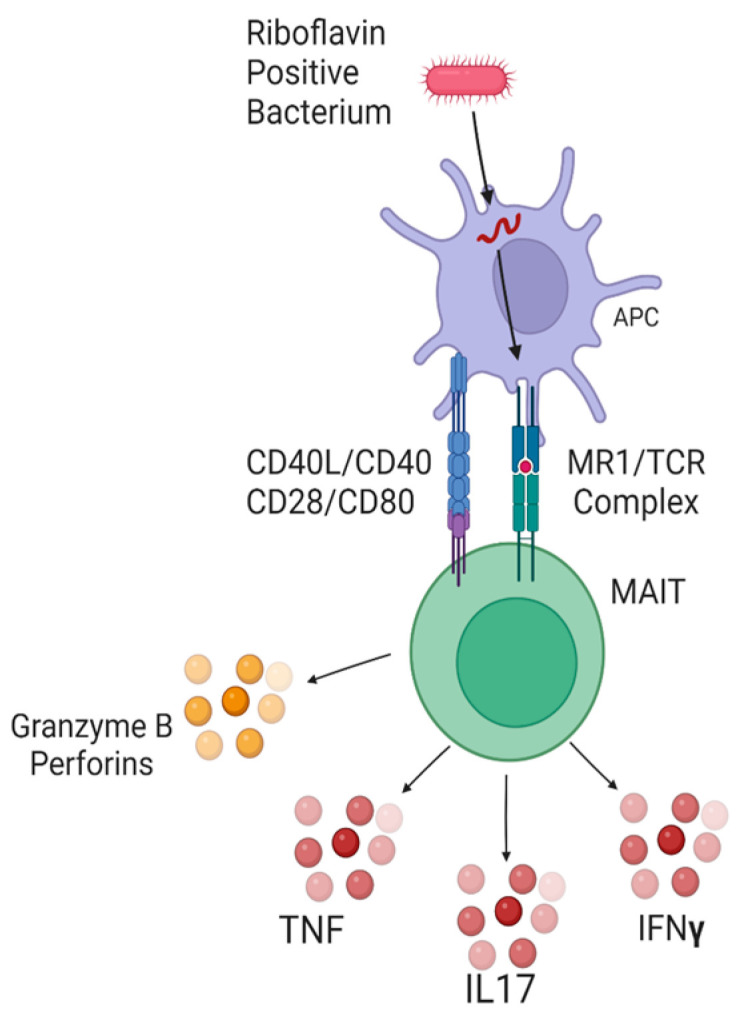
TCR-dependent activation of MAIT cells. The APC uptakes the bacterium and presents a riboflavin component on its MR1 complex. TCR-dependent activation requires APC co-stimulation through CD40/CD40L and CD28/CD80 interactions. Activated MAITs upregulate CD69 expression and produce TNF, IL17, IFNγ, and pro-apoptotic molecules.

**Figure 2 nutrients-16-01245-f002:**
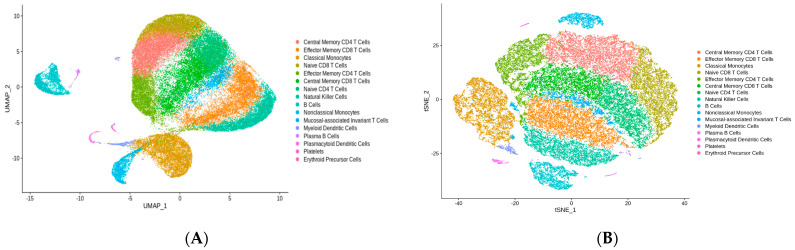
(**A**) UMAP depicting clustering result of 57,064 cells from 7 donors across 4 time points (n = 28 samples). Each point represents a single cell. Cluster color indicates cell type based on manual and automated annotation. (**B**) tSNE. (**C**) Feature plot with example markers used to identify MAITs. (**D**) Composition of cell types stratified by condition (*x*-axis).

**Figure 3 nutrients-16-01245-f003:**
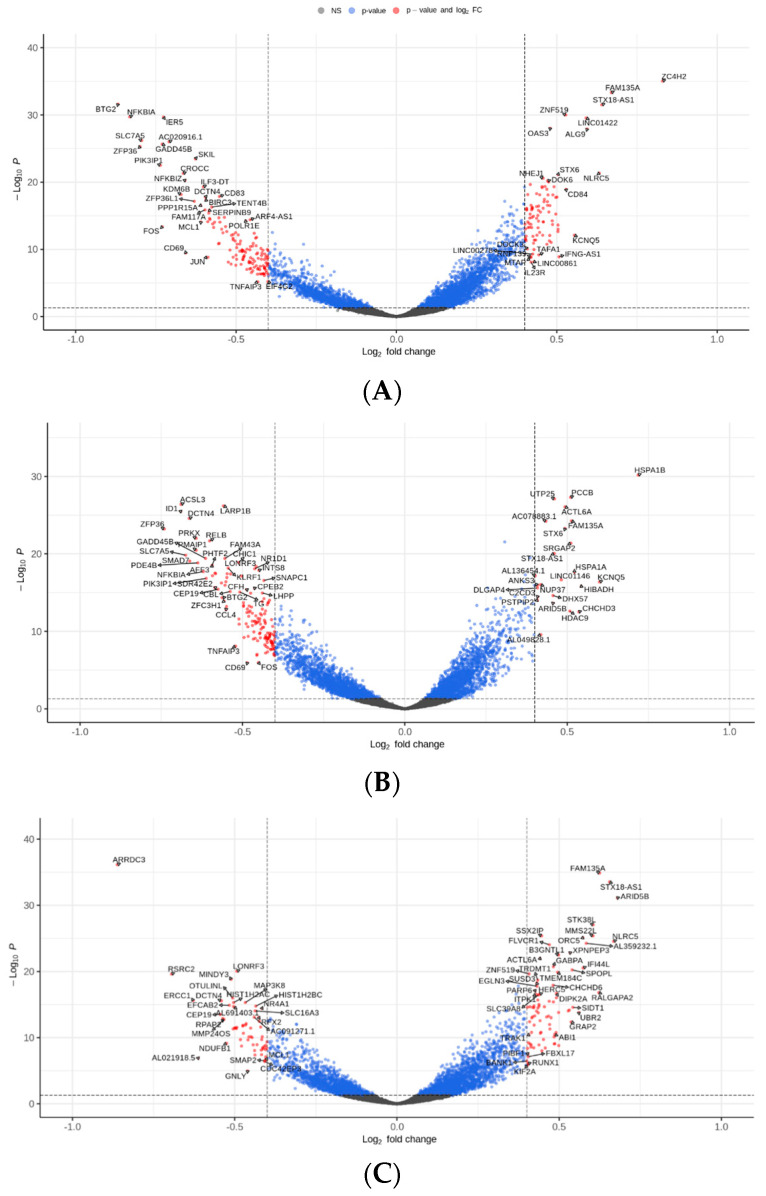
Differentially expressed genes (DEGs) compared to baseline after 2 (**A**), 4 (**B**), and 7 (**C**) days of dietary intervention. Red dots represent DEGs with an absolute LogFc ≥ 0.4 with FDR-adjusted *p*-value < 0.05.

**Figure 4 nutrients-16-01245-f004:**
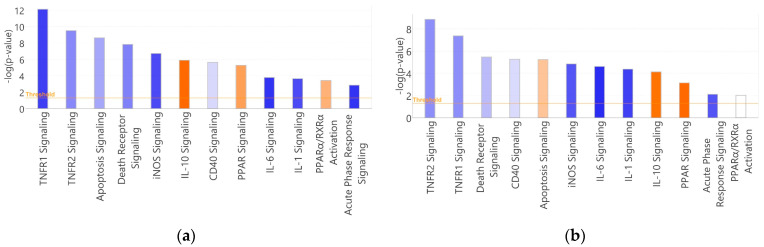
Select enriched canonical pathways on days 2 (**a**) and 4 (**b**) of dietary restriction. Threshold (1.3) indicates a -log Benjamini–Hochberg-corrected *p*-value of <0.05. Blue shading indicates the predicted inhibition (negative Z-score) of the pathway based on the input dataset. Orange shading indicates activation (positive Z-score). White shading indicates no conclusive evidence for activation or inhibition. Darker blue or orange hues indicate more significant inhibition or activation.

**Figure 5 nutrients-16-01245-f005:**
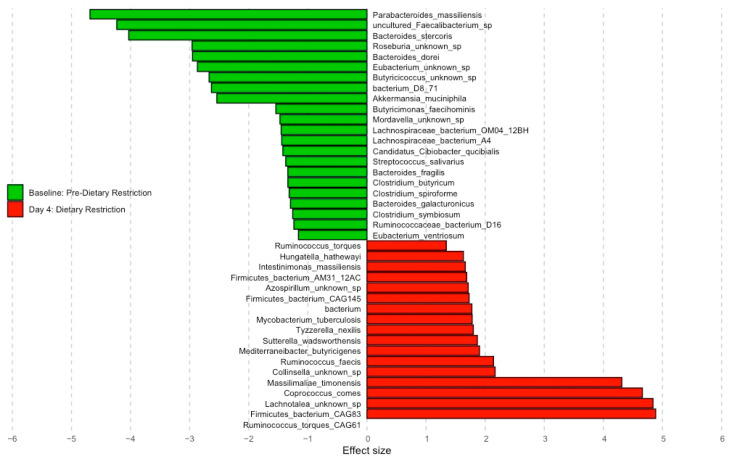
Effect size graph comparing species-level changes from baseline to day 4 of StDR (FDR *p* < 0.1, effect size > 1.0).

**Figure 6 nutrients-16-01245-f006:**
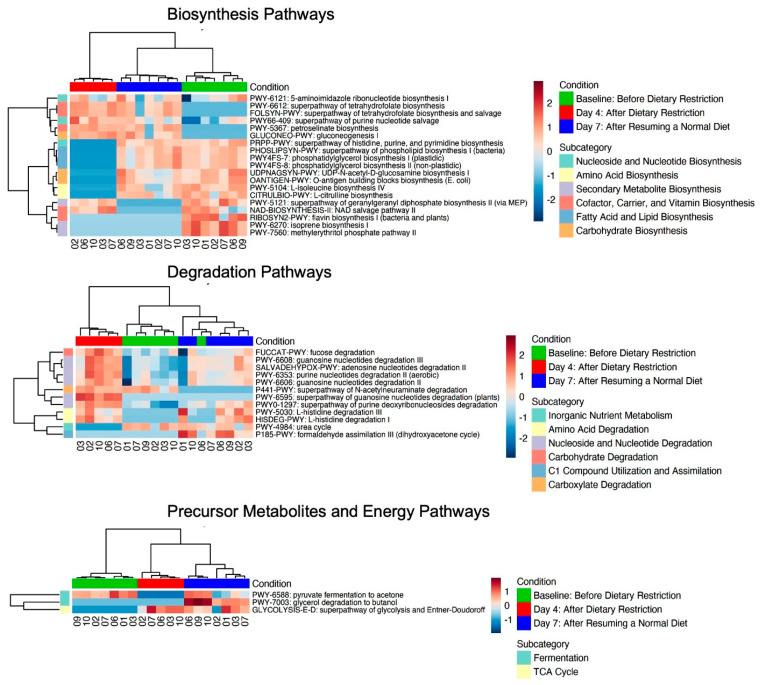
Heatmaps of significantly different metabolic pathways (FDR-adjusted *p* < 0.05). Pathways are classified as biosynthetic, degradation, and precursor metabolite and energy pathways according to MetaCyc ontology.

**Table 1 nutrients-16-01245-t001:** Participant characteristics.

Subject	Sex	Race/Ethnicity	Age	BMI	Medications	Tobacco Use	Daily Energy Requirement (kcal)
1	F	Asian	22	23.4	Oral Contraceptive	No	1940
2	M	White	30	28.9		No	3267
3	M	White	26	26.9	Magnesium, Krill Oil	No	2983
4	M	White	28	22.4		No	2884
5	F	Hispanic	23	32.2	Oral Contraceptive	No	2116
6	F	White	19	18.9	Oral Contraceptive	No	2221
7	M	White	29	27.1		No	3118

**Table 2 nutrients-16-01245-t002:** Top 10 most significantly affected regulatory chemokines and complexes after 2 and 4 days of StDR. All depicted regulators were significant based on Fisher’s exact test with a Benjamini–Hochberg-corrected *p*-value of overlap set to <0.01. The function describes the general action or purpose of each regulator. A negative Z-score < −2.0 predicts significant regulator inhibition. All of the most significant regulators were inhibited. Day 7 is not depicted, as no functionally significant regulators were identified compared to the baseline.

Day 2			Day 4		
Regulator	Function	Z-Score	Regulator	Function	Z-Score
CD3	TCR co-receptor	−2.1	CD3		−3.3
NFκB Complex	Cell stress response	−3.7	NFκB Complex		−4.4
CD40LG	Costimulation	−2.7	CD40LG		−3.2
GM-CSF	Proinflammatory cytokine	−2.2	GM-CSF		−3.4
IL2	T-cell growth	−3.0	IL2		−2.6
KLF6	Tumor suppressor gene	−3.1	KLF6		−3.1
TNF	Proinflammatory cytokine	−3.9	TCR	TCR complex	−2.5
IL1B	Proinflammatory cytokine protein	−3.6	RELA	NFκB complex subunit	−3.6
CD40	Costimulation	−2.2	CD28	Costimulation	−2.8
EPHA2	Protein kinase	−2.1	IFNγ	Proinflammatory cytokine	−2.5

## Data Availability

The original contributions presented in the study are included in the article and Appendix A, further inquiries can be directed to the corresponding author.

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
