# Peer review of "Short-Term Dietary Restriction Potentiates an Anti-Inflammatory Circulating Mucosal-Associated Invariant T-Cell Response"

_nutrients, 2024, doi:10.3390/nu16081245_

Round 1
Reviewer 1 Report
Comments and Suggestions for Authors
Scott et al. submitted the manuscript entitled: Short-term Dietary Restriction Potentiates an Anti-inflammatory Circulating Mucosal-associated Invariant T-cell Response, in which they investigated on influences of StDR on MAITs. The authors identified several targets and pathways related to immunoresponse in MAITs and also, discussed influences on microbials. In general, this is a well-written and informative manuscript and the topic will be of interest to potential readers of Nutrients.
My comments are as follows.
1. Page 8, figure 3: For the overlap of identified genes in day2/4 and day7, are they independent with StDR or previous StDr can still take effect after 3-day normal diet? The authors can also consider generating a table of identified genes that showed same expression trend between day 2 and day 4, but different in day 7.
2. Page 12, discussion: The authors stated that StDR may have the altered the microbiome, potentially producing favorable MAIT modulating 383 metabolites. This is a rational assumption but need more evidences. In this manuscript, although the authors analyzed species abundance and microbial functional pathway analysis, still not enough connections between microbial and MAIT. The authors are suggested to discuss more on these two and include related references.
Reviewer 2 Report
Comments and Suggestions for Authors
The study focuses on short-term dietary restriction examining the transcriptomic response of mucosal-associated invariant T cells (MAIT) using single-cell RNA sequencing of human PBMCs, while in parallel evaluating the gut microbial species-level changes in stool samples. The study is very interesting, well-performed and well-presented.
Some minor issues to be addressed are the following:
1. The authors should discuss the gut microbiota composition in relation to StDR in the Discussion section.
2. What about peripheral immune responses to StDR? I think there is relevant literature that could be discussed in the Discussion section. What about a general blood examination analysis ?
Comments on the Quality of English Language
Minor editing of English language is needed.
